# Updates on Aβ Processing by Hsp90, BRICHOS, and Newly Reported Distinctive Chaperones

**DOI:** 10.3390/biom14010016

**Published:** 2023-12-22

**Authors:** Mohammed Iqbal, Shea-Lorane Lewis, Shivani Padhye, Umesh Kumar Jinwal

**Affiliations:** Department of Pharmaceutical Sciences, USF-Health Taneja College of Pharmacy, University of South Florida, Tampa, FL 33612, USA; miqbal2@usf.edu (M.I.);

**Keywords:** Alzheimer’s disease, amyloid-β protein, Hsp90, BRICHOS domain chaperone, chemical chaperone, artificial chaperone, engineerable chaperone

## Abstract

Alzheimer’s disease (AD) is an extremely devastating neurodegenerative disease, and there is no cure for it. AD is specified as the misfolding and aggregation of amyloid-β protein (Aβ) and abnormalities in hyperphosphorylated tau protein. Current approaches to treat Alzheimer’s disease have had some success in slowing down the disease’s progression. However, attempts to find a cure have been largely unsuccessful, most likely due to the complexity associated with AD pathogenesis. Hence, a shift in focus to better understand the molecular mechanism of Aβ processing and to consider alternative options such as chaperone proteins seems promising. Chaperone proteins act as molecular caretakers to facilitate cellular homeostasis under standard conditions. Chaperone proteins like heat shock proteins (Hsps) serve a pivotal role in correctly folding amyloid peptides, inhibiting mitochondrial dysfunction, and peptide aggregation. For instance, Hsp90 plays a significant role in maintaining cellular homeostasis through its protein folding mechanisms. In this review, we analyze the most recent studies from 2020 to 2023 and provide updates on Aβ regulation by Hsp90, BRICHOS domain chaperone, and distinctive newly reported chaperones.

## 1. Introduction

Neurodegenerative diseases are a diverse group of neurological disorders that negatively impact the lives of millions of people across the world through the progressive loss of neurons [1]. Among the neurodegenerative diseases, Alzheimer’s disease (AD) is linked to the majority of dementia cases. AD was initially reported 100 years ago by a German psychiatrist named Alois Alzheimer when he observed fibrillary structures within the brains of patients that came to him with progressive cognitive dysfunction [2]. AD, along with other ageing diseases, results in progressive lifestyle and cognitive decline, profoundly impacting the quality of life and functional independence of about 55 million people worldwide [3]. The development of these diseases is attributed to the aggregation of misfolded proteins, which leads to progressive neuronal loss and subsequent dysfunction [4]. These aggregates are insoluble fibrils characterized by a β-pleated sheet usually arranged in an antiparallel configuration [5,6]. In AD, β-amyloid (Aβ) aggregates into different oligomeric forms and amyloid fibrils. These fibrils are a major part of amyloid plaques in the brains of AD patients [7]. Based on the involvement of Aβ aggregates in AD pathogenesis, they serve as a promising target for AD therapy [8].

The formation of protein aggregates is the result of failed protein quality control machinery, the purpose of which is to refold or degrade misfolded proteins [9]. A main component of the body’s quality control machinery is chaperone proteins. Molecular chaperones play a vital role in the body since they are the regulators of protein conformational states [9,10,11]. Hence, chaperone proteins and targeting medicinal agents pose great potential towards therapeutic intervention for AD and other related neurodegenerative diseases. The major molecular chaperone heat shock proteins (Hsps) are a family of stress proteins that play an important role in protein processing [12,13,14]. Heat shock protein 90 (Hsp90) is a crucial molecular chaperone that is responsible for many mechanistic and therapeutic pathways in various diseases. The first report by Yoo et al. highlighted the significance of the deranged expression of HSPs and various other chaperones in AD cases [15]. Furthermore, Hsp90 targeting compound 17-AAG has been shown to attenuate Aβ toxicity and prevent memory loss in AD models [16]. Recent reports suggest that the Dihydropyridine derivative LA1011 compound’s binding to Hsp90 in pathological condition leads to heat shock response and neuroprotection [17,18]. Very recently, Roe et.al. crystallized and characterized the Hsp90–LA1011 complex [19]. These newly reported data on LA1011 along with temperature-dependent Hsp90-mediated Aβ production and Hsp90-ATP-mediated Aβ fibrillation data (discussed in this review) provide further evidence on the importance of the Hsp90 chaperone in the development of AD treatment options. Therefore, chaperone activity is a main area of focus in the efforts to target Aβ aggregates.

The latest developments in AD research showed that the modulation of Hsp90, the BRICHOS domain, and several other prominent molecular chaperone proteins play a critical role in the regulation of Aβ processing. The name BRICHOS comes from BRI2, CHOndromodulin-I and Surfactant protein C. It is composed of 100 amino acid residues and has various fascinating properties including alleviating Aβ fibrillation and aggregation. Proteins that have this BRICHOS domain have been observed to have a diverse assortment of phenotypes including dementia, respiratory distress, chondrosarcoma, and stomach cancer [20]. This domain has been seen in 12 different protein families up until this point. All BRICHOS domain containing proteins and Buxbaum et al. provided here [21]. These proteins harbor similar predicted secondary structures [21]. The BRICHOS domain has also been linked to lung fibrosis, dementia, and cancer [22,23,24]. Artificial chaperones are chaperone–mimetic molecules that selectively associate with client substrates for proper folding and processing. These chaperones have been developed for clinical applications [25]. For example, Benzimidazole-functionalized polyfluorene (PFBZ) is an artificial chaperone that has been shown to prevent Aβ-mediated neuronal damage [26]. Chemical chaperones are low-molecular-weight small compounds that are associated with proteins and affect the denaturation and trafficking of proteins. Brown et al. and Sato et al. suggested the use of the chemical chaperone term for certain small molecules affecting the activity of disease-associated proteins [27,28]. The chemical chaperone 4-phenylbutyric acid (PBA) is an FDA-approved compound that has been shown to promote endoplasmic reticulum stress reduction [29]. PBA is also studied heavily in AD cellular and animal models [30,31,32]. Recently, Baumanns et al. and Villani et al. reported PBA-mediated Aβ regulation [33] and PBA kinetics [34], respectively. The plant compound alkannin was reported to act as a chaperone in Aβ aggregation inhibition [35]. This compound also has been proposed as an antiaging agent [36]. In this review, we focus on chaperone-mediated Aβ processing in AD. Specifically, we highlight updates from the last 3 years (2020–2023) on Hsp90, BRICHOS, artificial chaperones, chemical chaperones, and molecular chaperones with engineerable protein scaffolds.

## 2. Hsp90

### 2.1. Hsp90–LA1011 Complex 

Hsp90 is capable of both solubilizing and suppressing protein aggregates and also the most prevalent chaperones in mechanistic pathways. Several reports suggested that Hsp90-targeting compounds represent great potential for the development of AD therapy. Recently, Hsp co-inducer 1,4-dihydropyridine derivative LA1011 was shown to have a neuroprotective effect [17]. One of these major discoveries lies in LA1011 complex formation and explores how its structure works with various co-chaperones of Hsp90: FKBP51, FKBP52, PP5, and CHIP [19]. By forming complexes with Hsp90, FKBP51 can influence the responsiveness of these receptors to steroid hormones, which are pivotal in regulating immune responses, inflammation, and cellular stress [37]. FKBP51 modulates the conformation and activity of specific client proteins like the glucocorticoid receptor (GR) and regulates various kinases cascades including insulin signaling [37]. Comorbidities like diabetes are recognized to exacerbate Aβ plaques in the context of age-related disorders. Inhibiting insulin binding impairs the brain’s synaptic plasticity, neuroinflammation, and the brain’s ability to metabolize glucose. A decrease in synaptic plasticity in AD individuals is one of the more noticeable ailments since it represents cognitive decline. Also, as a result of a decrease in synaptic plasticity, neuroinflammation activates and promotes tau accumulation [38]. Increased levels of FKBP51 in conjunction with age and stress are seen to rise in individuals with AD [19]. In transgenic mice models, LA1011 was shown to disturb the binding of FKBP51 to allow for Hsp90 to reduce disease-causing agents [19]. The results of the study indicate that there is competition for binding to the hydrophobic pocket at the C-terminal end of Hsp90 between the helical extension of FKBP51’s TPR domain and LA1011 [19] Table 1, Figure 1. The dihydropyridine LA1011 complex was also shown to decrease tau protein aggregation and Aβ plaque aggregation possibly via Hsp90 interactions in AD mice [17,19].

### 2.2. Temperature on Hsp90 Activity

Extremely high temperatures can induce cellular stress, which in turn can affect the expression and activity of HSP90 and other heat shock proteins. These proteins help cells cope with stress by preventing protein damage and aggregation. Observations revealed that sleep deprivation led to a rise in core body temperature, suggesting a connection between sleep deprivation and Aβ production in AD [39]. Inflammatory environments in individuals with AD create a domino effect in the Hsps’ response to antagonistic environments; inflammation downregulates Hsp production since the cell is incapable of managing the stress, increasing Aβ aggregates [38]. In disease conditions, Hsp90 has been shown to play a key role in aggregated protein processing and homeostasis. Slight changes in temperature from 37 °C to 39 °C led to enhanced Aβ40 and Aβ42 production in cell culture experiments [39]. Hsp90 knockdown experiments at higher temperatures showed increased γ-secretase complex formation and the abolishment of increased Aβ production. Mouse model studies conducted at standard and higher temperatures showed that higher temperatures resulted in increased Hsp90, PS1-CTF, NCT, and γ-secretase complex levels. This study proposed that elevated core body temperature resulted in heightened Aβ aggregation (Table 1, Figure 1) [39]. This observation points towards a novel mechanism of Aβ processing at elevated temperatures via the involvement of HSP90 in the formation of the γ-secretase complex.

### 2.3. Adenosine Triphosphate (ATP)-Mediated Changes in Hsp90 Activity

Neurons’ proper function relies on the efficient production of energy by the mitochondria. Mitochondrial dysfunction leads to reduced ATP generation, impairing neuronal activity and contributing to the cognitive decline observed in AD. Mitochondrial dysfunction is a significant contributor to the pathogenesis of AD and other neurodegenerative diseases, producing higher levels of reactive oxygen species (ROS) and NO, leading to oxidative stress [40,41]. Elevated ROS levels can cause damage to mitochondrial components, including lipids, proteins, and DNA, expediting impaired mitochondrial function. This damage can disrupt the electron transport chain, reduce ATP production, and compromise mitochondrial integrity, contributing to a vicious cycle of increasing ROS production and worsening mitochondrial dysfunction. Enhancing mitochondrial health could alleviate several aspects of AD progression and provide a multidimensional strategy to address the complex nature of the disease [42,43,44]. The Hsp90 chaperone machinery is an ATP-dependent process. Essentially, ATP prevents Hsp90 chaperones from performing their inhibitory mechanisms, indicating that chaperones can be manipulated [45,46]. Hsp90 undergoes conformational changes as it interacts with client proteins. These changes are fundamental for the proper folding and stabilization of client proteins [47,48]. ATP hydrolysis provides the energy needed to facilitate these conformational changes. According to a recent study, ATP reduces Hsp90′s inhibitory impact on Aβ40 fibrillation by decreasing the hydrophobic surface of Hsp90 [45]. When Hsp90 proteins are present, there is no indication of secondary β-sheet structure conformation, and the signal associated with the initial secondary Aβ40 random coil structure gradually diminishes over time, ultimately disappearing as incubation continues possibly due to Hsp90 and Aβ40 large aggregates forming (Table 1, Figure 1) [45]. These findings suggest that Hsp90′s presence keeps Aβ mostly in a monomeric form. Further studies are needed to confirm the effect Hsp90 and ATP have on Aβ fibrillization in AD and other neurodegenerative diseases.

**Figure 1 biomolecules-14-00016-f001:**
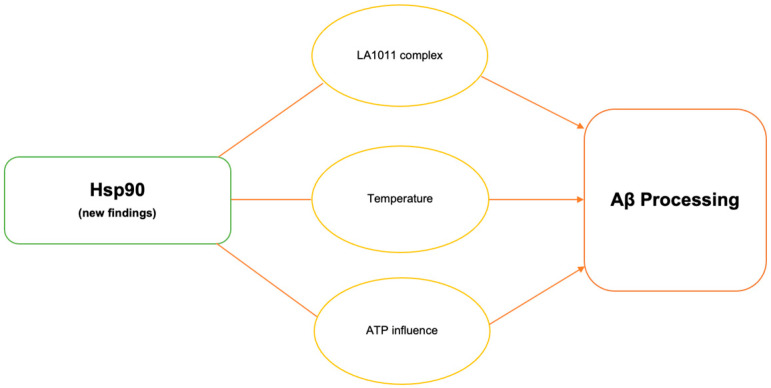
Schematic illustration of recent reports highlighting the importance of HSP90 in Aβ processing.

**Table 1 biomolecules-14-00016-t001:** Recent findings on HSP90mediated Aβ regulation.

Recent HSP90 Reports	Findings	Model Used	Reference
Hsp90–LA1011 complex	Crystal structure of the Hsp90–LA1011 complex	Recombinant protein	[19]
Influence of temperature on Hsp90 activity	HSP90-mediated Amyloid-beta production is heat-dependent	Cell model	[39]
ATP mediated changes in Hsp90 activity	ATP affects HSP90s ability to inhibit Aβ fibrillization	Recombinant protein	[45]

## 3. BRICHOS Chaperone Domain

### 3.1. BRICHOS Interactions

A chaperone that is due for major discussion is the BRICHOS chaperone domain. As shown in Figure 2, all BRICHOS-containing proteins have an N-terminal cytosolic segment, a hydrophobic transmembrane or a signal peptide region, a linker segment, and the BRICHOS domain. The C-terminal segment is present in all BRICHOS except proSP-C [21]. Recent studies have shown this chaperone’s involvement in Aβ processing in various ways (Table 2, Figure 3). Österlund et al. (2022) used structure prediction algorithm AlphaFold2 and mass spectrometry (MS) to define the BRICHOS domain’s interaction with Aβ. They used the BRICHOS-domain containing proform of lung surfactant protein C (proSP-C) for X-ray crystallography [49]. This proSP-C protein is reported to inhibit AB fibrilization [50]. Furthermore, Shimozawa et al. showed that BRICHOS interacts with multiple proteins [51]. They used mice brain slice cultures incubated with recombinant BRICHOS Bri2 to perform proteomic analysis. They detected the binding of the Spectrin alpha chain, Spectrin beta chain, Myosin-10, unconventional Myosin-Va, Drebrin, Tubulin beta-2A chain, and Actin cytoplasmic 1 to BRICHOS Bri2. However, in this study, there was no interaction between Aβ and BRICHOS Bri2 observed, possibly due to the already formed Aβ aggregates in plaques in this brain slice culture model. Overall, Shimozawa’s study suggests that BRICHOS Bri2 may play an important role in cytoskeleton regulation.

### 3.2. S100A9 and Bri2 BRICHOS

Another recent study by Manchanda et al. (2023) generated a mutant form of recombinant human Bri2 BRICHOS R221E and tested it in AD animal models [52]. Their findings suggest that the mutant form is more efficient than the wild-type form in preventing Aβ42-mediated toxicity in mouse slice culture. Furthermore, using the mouse model, they also showed that the mutated recombinant form is BBB-permeable. They also performed behavioral testing after treating an AD mouse model with the mutated Bri2 BRICHOS when AD pathology began to be observable through symptoms. The behavioral testing data showed that mutated Bri2 BRICHOS helped in boosting recognition and working memory, as was observed through an object recognition test. It is important to note that these positive results were only seen if the treatment started as soon as AD pathology began because this treatment did not show any improved symptoms if it was delayed by 4 months. To confirm these results, the amount of mutated Bri2 BRICHOS found in the brain correlated with the improvement and regaining of recognition and memory [52].

Andrade-Talavera et al. investigated the effect of human recombinant Bri2 BRICHOS on S100A9 amyloid kinetics [53]. S100A9 was reported to contribute to amyloid processing and neuroinflammation in AD, Parkinson’s disease, and traumatic brain injury. It forms an intracellular amyloid oligomer and Aβ co-aggregates in AD [54,55,56]. S100A9 is an amyloidogenic polypeptide that is inhibited by recombinant Bri2 BRICHOS. The BRICHOS chaperone is seen to cap amyloid fibrils, which not only reduces aggregation but also decreases the inflammatory response (Table 2, Figure 3) [53].

### 3.3. BRICHOS Domain Dependent on Conserved Asp Residue

While all of these recent developments are noteworthy and important, Chen et al.’s data on the conserved aspartate (D) residue in the BRICHOS domain provides further evidence of the potential of this chaperone [57]. This particular residue has been consistent across most BRICHOS domain-containing proteins. A few BRICHOS domain-containing proteins have an aspartate to asparagine (N) mutation. Human Bri2 BRICHOS D148N mutation promotes dimerization without affecting function. Their findings suggest that the conserved aspartate residue’s pKa value ranges from pH 6.0 to 7.0, and due to its ionized state, aspartate promotes the domain’s structural flexibility (Table 2, Figure 3) [57]. This chaperone’s role in the context of Hsp90 in Aβ processing is yet to be defined. Hence, future studies will help understand the role of the Bri2 BRICHOS chaperone in AD and its therapeutic potential.

**Table 2 biomolecules-14-00016-t002:** Recent findings on BRICHOS chaperone domain-mediated Aβ regulation.

Recent BRICHOS Chaperone Domain Reports	Findings	Model Used	Reference
BRICHOS interactions	Hydrophobic interactions between BRICHOS chaperone domain and amyloid-beta	Recombinant protein	[49]
S100A9 and Bri2 BRICHOS	S100A9 is inhibited by recombinant Bri2 BRICHOS	Mouse brain slice culture, recombinant protein	[53]
Bri3 BRICHOS domain chaperone’s efficiency	Bri3 BRICHOS domain is more effective in preventing non-fibrillar protein aggregation	Recombinant protein	[23]
BRICHOS domain dependent on conserved Asp residue	Asp residue is required for the flexible state of BRICHOS	Recombinant protein	[57]

### 3.4. Bri3 BRICHOS Domain Chaperone’s Efficiency

Poska et al. compared the efficiency of BRICHOS Bri2 and Bri3 against Aβ fibrillar and non-fibrillar aggregation [23]. Bri2 and Bri3 BRICHOS domains have approximately 60% identical sequences, which indicates similarities in their structure and functions. Bri2 and Bri3 involvement in Aβ has been previously reported [58,59]. A recent report from Poska et al. suggests that Bri2 and Bri3 BRICHOS have differences in regulating Aβ. Bri2 was found to be more potent against Aβ fibrillar aggregation whereas Bri3 was more potent against non-fibrillar aggregation (Table 2, Figure 3). Bri3 is only expressed in the CNS whereas Bri2 is expressed in the CNS as well as peripheral tissues [60,61]. Overall, Bri2 and Bri3 have different tissue distributions and capacities as molecular chaperones. Future studies using mutants and AD models will help in fully understanding their involvement in AD pathogenesis.

## 4. Artificial Chaperone Benzimidazole-Functionalized Polyfluorene

Benzimidazole-functionalized polyfluorene (PFBZ) is a newly reported artificially synthesized chaperone molecule for preventing Aβ-mediated neuronal damage. PFBZ inhibits Aβ aggregate interaction with the cell membrane by wrapping around Aβ, which eventually leads to reduced Aβ-mediated cytotoxicity [26]. Benzimidazole derivatives have been investigated for their ability to inhibit or disrupt the aggregation of Aβ peptides into toxic forms, such as amyloid plaques [62,63]. PFBZ was purposefully engineered to contain a part of Thioflavin T (ThT), a known amyloid sequester in the benzimidazole group [26]. This engineered PFBZ chaperone has nanodimensions (a size of 9.04 ± 1.58 nm), giving it a blood brain barrier (BBB) permeability feature and bright yellow emission for imaging. The use of a ThT assay to monitor PFBZ’s effect on AB fibrillation showed that increasing PFBZ concentrations completely inhibited Aβ fibrillation, showing no ThT signal detection [26]. PFBZ was also shown to reduce mitochondrial disfunction via affecting Aβ-mediated ROS production. In vivo, PFBZ has BBB permeability, thus protecting the brain from amyloid-mediated neuroinflammation [26] (Table 3, Figure 4). Aβ deposition in blood vessels has been shown to trigger brain hemorrhage [64]. Mondal et al. used wild-type mice injected with Aβ in the tail veinfor testing the effect of PFBZ on hemorrhages. Their data suggest that PFBZ treatment recovers mice from Aβ-mediated hemorrhages [26]. Overall, PFBZ possesses unique properties for developing novel AD treatments. Further validation studies using AD transgenic animal models might strengthen its potential for therapeutic use.

## 5. Molecular Chaperone with Engineerable Protein Scaffolds

The 43 kD subunit of the chloroplast signal recognition particle (cpSRP43) is an ATP-independent plant chaperone shown to target Aβ oligomers [65]. It contains an engineerable protein scaffold ankyrin repeat domain (ARD). ARDs are a highly abundant protein motif in nature and are composed of 30–40 amino acid residues [66]. They almost exclusively function to moderate protein–protein interaction, and some can potentially result in the occurrence of human disease [66]. Generally, four to six copies of the ankyrin repeat stack on top of each other, forming an elongated structure that has a hydrophobic core throughout it [67]. Each repeat forms two antiparallel a-helices antiparallel to one another to make a helix–turn–helix conformation [65,66]. A loop extends at a 90-degree angle from the structure in order to facilitate the formation of a hairpin-like b-sheet with neighboring motifs [66]. Consensus sequences both within and in between stacked repeats work to stabilize the ARDs while numerous residues project from the protein surface, providing binding sites for specific protein partners. Using this information, ARDs can be designed to have high specificity and affinity for a number of target proteins [65]. In the case of cpSRP43, biochemical analysis showed that its substrate-binding domain had high anti-amyloid activity against various forms of Aβ such as Aβ42, Aβ40, and ASyn. Fibrillation of Aβ42 was significantly slowed by cpSRP43 in a dose-dependent manner [65]. Transmission electron microscopy (TEM) imaging showed that Aβ40 and A syn fibrillation were significantly delayed in the presence of cpSRP43. The mechanism through which cpSRP43 inhibits Aβ42 and Aβ40 aggregation is the inhibition of secondary nucleation, which was found using the kinetic analysis pipeline [65,68]. To analyze which structural domain of CpSRP43 is involved in anti-fibril activity, Gupta et al. generated deletion constructs for testing. Their results showed that the substrate-binding domain (SBD) of CpSRP43 is crucial for its anti-fibril activity [65]. Furthermore, they analyzed full-length CpSRP43 and an SBD-deleted construct for neuroprotection against Aβ toxicity using neuroblastoma SH-SY5Y cells and the CellTiterGlo assay. Their data suggest that CpSRP43, particularly its SBD domain, is necessary for protection against Aβ toxicity in neuroblastoma cells (Table 3, Figure 4). Taking all of this together, cpSRP43 is an excellent candidate for bioengineering to generate client protein-specific enhanced chaperone activity [65].

## 6. Chemical Chaperones

### 6.1. Alkannin

Chemical chaperones have been shown to target aggregated proteins in cystic fibrosis and other diseases [69,70,71]. Recently, Hosoi et al. performed chaperone activity and Aβ aggregation measuring assays on Kampo medicine compounds to find viable and safe Aβ-targeting chemical chaperones for AD treatment. Their screening data showed that the plant compound alkannin acts as a chaperone to inhibit Aβ aggregation [35]. They found that alkannin and its enantiomer shokonin exhibited chaperone activity as well as inhibitory effects on Aβ aggregation [35]. It was isolated from the roots of alkannin tinctoria and it has been shown to have wound healing and anti-tumor properties [72,73]. Using circular dichroism (CD) spectral analysis, Hosoi et al. showed that after 24 h of incubation with alkannin and Aβ, the CD structure was ameliorated. These findings suggest that alkannin affects the β-sheet structure, which may eventually cause changes in Aβ conformation [35]. Furthermore, electron microscopy analysis showed that the alkannin-mediated inhibition of Aβ fibril formation was moderately affected. They also performed an in vitro test using PC12 neuronal cells to find out whether or not alkannin has any neuroprotective effect. The test results showed that alkannin protects cells from Aβ-mediated toxicity. Further characterization of alkannin’s effect using a *C. elegans* AD model showed that alkannin treatment recovers the chemotaxis phenomenon. A ThT assay demonstrated alkannin reduces Aβ aggregation in this model (Table 3, Figure 4) [35]. Overall, Hosoi et al. showed the potential of alkannin as a neuroprotective agent for AD treatment.

### 6.2. 4-Phenylbutyric Acid

The FDA-approved compound 4-Phenylbutyric acid (PBA) has been shown to act as chemical chaperone as well as an ammonia scavenger and an inhibitor of histone deacetylase [31,32,33]. It is a short-chain fatty acid compound that is used in the treatment of urea cycle disorders [32]. PBA has also been investigated as an AD treatment option using APPswe/PS1delta9 AD transgenic mice. Treatment with PBA in AD mice was shown to improve memory retention possibly via amyloid plaque reduction in the cortex and hippocampus [30]. Recently, Baumanns et al. investigated this compound using a *C. elegans* AD model expressing human Aβ. They showed a reduction in Aβ aggregation in PBA-treated *C. elegans* and an improvement in motility. RNAi knockdown data for heat shock factor-1 (HSF-1) was shown to block the PBA effect in this model. These data suggest that PBA activity is dependent on HSF-1 and the activation of the heat shock response in the *C. elegans* AD model [33]. Furthermore, PBA treatment in 3xTg-AD astrocytes has been shown to improve astrocytic protein synthesis [74]. Very recently, Villani et al. developed a liquid chromatography high-resolution mass spectrometry technique for PBA quantification [34]. They used this technique to estimate the absorption and adsorption kinetics of PBA in AD astrocyte cell models as well as melanoma cell lines (Table 3, Figure 4). Additionally, they correlated the concentration of PBA with the recovery of protein synthesis in AD astrocytes. Overall, these studies demonstrate the importance of chemical chaperones such as PBA in AD research and therapeutic development.

## 7. Conclusions

Neurodegenerative diseases have a complex pathogenesis with multiple means of dysfunction. A main characteristic of this group of disorders is the misfolding of proteins as a result of inadequate activity from chaperones in the body. The misfolding of Aβ often leads to the formation of aggregates in many neurodegenerative diseases such as AD. Because of this, novel therapies for neurological disorders revolve around proteins that have the ability to eliminate these aggregates, such as chaperones. Molecular chaperones like Hsp90 assist in the correct folding and stabilization of proteins, including Aβ. Modulation of Hsp90 activity and interactions have been shown to affect Aβ processing. Recent findings on Hsp90–LA1011 complex formation, the effect of temperature, and the presence of ATP further emphasize its role in Aβ regulation. New developments on the BRICHOS chaperone domain further characterize its involvement in Aβ processing. The main action of BRICHOS was seen to reduce Aβ fibrillation and aggregation. This is critical since it is the hallmark of AD. Recent studies highlighted different interactions that this domain had with Aβ, such as hydrophobic interactions, Aβ capping, etc., that allow for regulation of Aβ. Studies on recently developed artificial PFBZ, chemical alkannin and PBA, and potentially engineerable cpSRP43 chaperones have been shown to be successful in clearing Aβ aggregates (Figure 1). They present as strong candidates for future studies. Overall, the findings discussed in this review emphasize the critical role of chaperones in Aβ processing and highlight the great potential for these chaperones in mechanistic studies as well as in the development of novel AD therapies.

## Figures and Tables

**Figure 2 biomolecules-14-00016-f002:**
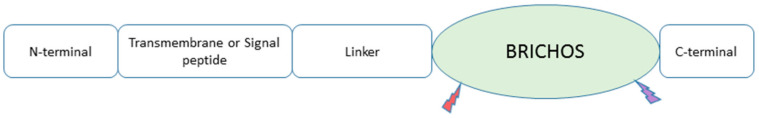
A model illustrating the BRICHOS domain-containing protein. It contains an N-terminal cytosolic segment, a transmembrane or a signal peptide region, a linker segment, the BRICHOS domain, and a C-terminal segment. Mutations in Bri2 BRICHOS are shown as tiny lightning bolt shapes; red represents the D148N mutation and purple represents the R221E mutation.

**Figure 3 biomolecules-14-00016-f003:**
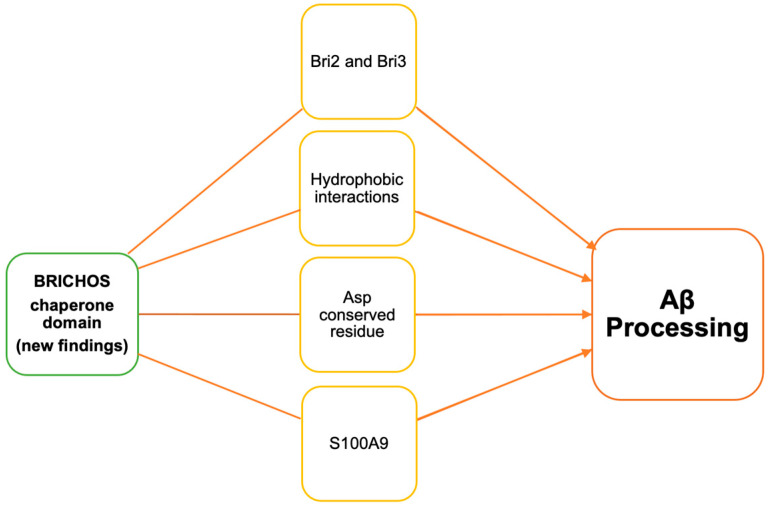
Schematic illustration of recent reports on BRICHOS domain-containing proteins involved in Aβ processing.

**Figure 4 biomolecules-14-00016-f004:**
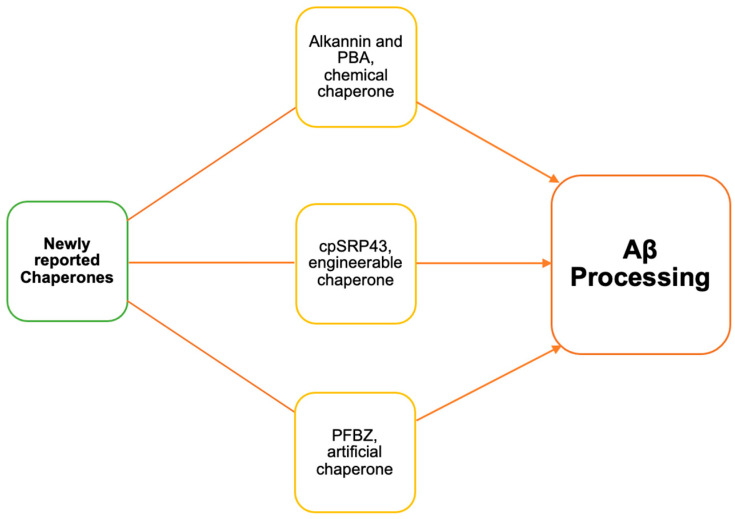
Schematic illustration of newly reported chemical engineerable and artificial chaperones in Aβ processing.

**Table 3 biomolecules-14-00016-t003:** Recent findings on novel chaperones involved in Aβ processing.

Novel Chaperones Targeting Aβ	Findings	Model Used	Reference
Artificial Chaperone	Inhibition of Aβ fibrillization, BBB permeability	Recombinant protein, cells, mouse	[26]
Molecular chaperone with engineerable protein scaffolds	Bioengineerable ARD domain containing cpSRP43 inhibits Aβ42 and Aβ40 aggregation	Recombinant protein, cells	[65]
Chemical Chaperones: Alkannin PBA	Alkannin reduces Aβ aggregation PBA-mediated Aβ reduction depends on HSF-1 PBA absorption and adsorption kinetics estimation	Recombinant protein, cells, *C. elegans* *C. elegans*, cells Cells	[35] [33] [34]

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
