# Peer review of "Updates on Aβ Processing by Hsp90, BRICHOS, and Newly Reported Distinctive Chaperones"

_biomolecules, 2023, doi:10.3390/biom14010016_

Round 1

Reviewer 1 Report

Comments and Suggestions for Authors

In the paper titled "Updates on Aβ Processing by Hsp90, BRICHOS, and Newly Reported Distinctive Chaperones" authored by Mohammed Iqbal et al., a comprehensive summary of recent studies surrounding Hsp90, BRICHOS, and various chaperones associated with Aβ processing is presented.

Alzheimer's disease is a devastating neurodegenerative condition characterized by the misfolding and aggregation of Aβ alongside abnormalities in hyperphosphorylated tau protein. While current strategies have shown some success in slowing the disease's progression, discovering a definitive cure remains challenging due to the complexity of Alzheimer's pathogenesis. Consequently, delving into the molecular mechanisms of Aβ processing and exploring alternative options, such as chaperone proteins like HSP90, holds significant promise.

BRICHOS domain chaperones play a pivotal role in maintaining cellular homeostasis by facilitating the correct folding of amyloid peptides and inhibiting their aggregation. Recent research has shed light on the involvement of artificial chaperones, chemical chaperones, and molecular chaperones possessing engineerable protein scaffolds in Aβ processing in Alzheimer's disease.

While the review is well-structured and coherent, several enhancements could be made once the authors address my concerns.

  1. 1. Add new figures for each section, like the protein organization of the Hsp90-LA1011 Complex in section 2.1, enhancing clarity and accessibility of information.
  2.  
  3. 2. Ensure references are provided where lacking, such as Line 104.
  4.  
  5. 3. Clarify the link between mechanisms and AD pathology for better understanding, especially in Line 122.
  6.  
  7. 4. Incorporate descriptions regarding the BRICHOS domain's features and proteins containing this domain, possibly via a model presenting the BRICHOS domain and a schematic illustrating protein domains with mutation labeling of Bri2 to elucidate their effects.
  8.  
  9. 5. Include conserved sequence alignment discussions, particularly addressing the role of the Asp residue in BRICHOS functions.
  10.  
  11. 6. Integrate other pertinent advancements like "Deranged Expression of Molecular Chaperones in Brains of Patients with Alzheimer's Disease."
  12.  
  13. 7. Consider creating a figure illustrating potential mechanisms of chaperones in AD progression to aid rapid comprehension, rather than relying solely on Figure 1.

Additionally, the paper could benefit from the inclusion of the following missing references:

  • 1. "Recombinant Bri3 BRICHOS domain is a molecular chaperone with effect against amyloid formation and non-fibrillar protein aggregation."
  •  
  • 2. "A new kid in the folding funnel: Molecular chaperone activities of the BRICHOS domain."

Author Response

We would like to thank the reviewers for providing excellent suggestions for improving our manuscript. Please see below the reviewers’ comments in Bold and our responses in regular fonts. Let us know if any additional information or changes are needed.  

Reviewer-1

While the review is well-structured and coherent, several enhancements could be made once the authors address my concerns.

  1. Add new figures for each section, like the protein organization of the Hsp90-LA1011 Complex in section 2.1, enhancing the clarity and accessibility of information. We have added figures in each section as suggested. In the revised manuscript, we have a total of four figures.
  2. Ensure references are provided where lacking, such as Line 104. We have added references as suggested in the revised manuscript in line 113 (which was originally line 104 in the first submission).
  3. Clarify the link between mechanisms and AD pathology for better understanding, especially in Line 122. We have added references as suggested in the revised manuscript in line 130 (which was originally line 122 in the first submission).
  4. Incorporate descriptions regarding the BRICHOS domain's features and proteins containing this domain, possibly via zb and a schematic illustrating protein domains with mutation labeling of Bri2 to elucidate their effects. We have included it in the introduction and the BRICHOS section. We have also added a new Figure 2 for the BRICHOS domain and mutations.
  5. Include conserved sequence alignment discussions, particularly addressing the role of the Asp residue in BRICHOS functions. We have included it in the introduction line numbers 64-69, with new references.
  6. Integrate other pertinent advancements like "Deranged Expression of Molecular Chaperones in Brains of Patients with Alzheimer's Disease." We have included it in the introduction in lines 49-51 with a reference.
  7. Consider creating a figure illustrating potential mechanisms of chaperones in AD progression to aid rapid comprehension, rather than relying solely on Figure 1. In the revised manuscript, we have multiple figures, references, and new information; hence, we feel these will be sufficient to provide details about this mechanism-related figure comment.

Additionally, the paper could benefit from the inclusion of the following missing references: we have updated references to include these.

  1. "Recombinant Bri3 BRICHOS domain is a molecular chaperone with effect against amyloid formation and non-fibrillar protein aggregation."
  2. "A new kid in the folding funnel: Molecular chaperone activities of the BRICHOS domain."

Reviewer 2 Report

Comments and Suggestions for Authors

In this manuscript, the authors give an overview of the roles of both natural and chemical chaperones in altering the production and aggregation processes of amyloid beta peptides.

While this review has quite substantial references, I'm not sure that the link between "natural" and "chemical" chaperones is there, particularly in the latter where they really just seem to be small molecules inhibiting aggregation rather than actually achieving "chaperone" function i.e. assist in refolding/unfolding. For the two examples of alkannin and 4-Phenylbutyric acid, do the authors have information with regards to them achieving chaperone function? Else I may suggest relabelling them or including the understanding that these are mostly small molecules inhibiting aggregation instead.

Specific minor comments:

Line 35: These aggregations are often formed within β-sheet rich protein fibrillations commonly 35 known as amyloid- β (Aβ) --> unsure what authors mean by this

Line 36: The process of amyloidosis refers to interactions between these fibrillary proteins which leads to aggregate formation --> unsure what authors mean by fibrillary proteins, perhaps referring to aggregation-prone proteins?

Line 57: Therefore, chaperone activity is a main area of focus in the efforts to combat Aβ aggregates --> do authors mean to target Ab aggregation? 

Comments on the Quality of English Language

I would also invite authors to recheck their manuscript for language errors. There is a tendency for the authors to use more colloquial words like "combatting" and wrong contractions like "who's" (line 317) which are not suitable for manuscripts.

Author Response

We would like to thank the reviewers for providing excellent suggestions for improving our manuscript. Please see below the reviewers’ comments in Bold and our responses in regular fonts. Let us know if any additional information or changes are needed.  

Reviewer-2

While this review has quite substantial references, I'm not sure that the link between "natural" and "chemical" chaperones is there, particularly in the latter where they really just seem to be small molecules inhibiting aggregation rather than actually achieving "chaperone" function i.e. assist in refolding/unfolding. For the two examples of alkannin and 4-Phenylbutyric acid, do the authors have information with regards to them achieving chaperone function? Else I may suggest relabelling them or including the understanding that these are mostly small molecules inhibiting aggregation instead.

We have updated the introduction section to clarify the chemical chaperone term use with additional references in the revised manuscript.  

Specific minor comments:

Line 35: These aggregations are often formed within β-sheet rich protein fibrillations commonly 35 known as amyloid- β (Aβ) --> unsure what authors mean by this. We apologize for this oversight. We have updated it.

Line 36: The process of amyloidosis refers to interactions between these fibrillary proteins which leads to aggregate formation --> unsure what authors mean by fibrillary proteins, perhaps referring to aggregation-prone proteins? We apologize for this oversight. We have updated it.

Line 57: Therefore, chaperone activity is a main area of focus in the efforts to combat Aβ aggregates --> do authors mean to target Ab aggregation?  We have updated it as suggested (in revised line #59).

Comments on the Quality of English Language

I would also invite authors to recheck their manuscript for language errors. There is a tendency for the authors to use more colloquial words like "combatting" and wrong contractions like "who's" (line 317) which are not suitable for manuscripts. We apologize for this. We have updated the text accordingly by substituting these words.